# An Overview of Sustainability Assessment Frameworks in Agriculture

Abdallah Alaoui [1,*,†], Lúcia Barão [2,†], Carla S. S. Ferreira [3,4,5] and Rudi Hessel [6]

1 Institute of Geography, University of Bern, Hallerstrasse 12, 3012 Bern, Switzerland
2 Center for Ecology, Evolution and Environmental Changes (cE3c), University of Lisbon, Campo Grande, 1749-016 Lisbon, Portugal; albarao@fc.ul.pt
3 Department of Physical Geography and Bolin Centre for Climate Research, Stockholm University, 11419 Stockholm, Sweden; carla.ferreira@natgeo.su.se
4 Navarino Environmental Observatory, Costa Navarino, Navarino Dunes, 24001 Messinia, Greece
5 Research Centre of Natural Resources, Environment and Society (CERNAS), Polytechnic Institute of Coimbra, Coimbra Agrarian Technical School, Bencanta, 3045-601 Coimbra, Portugal
6 Wageningen Environmental Research, 6708 WG Wageningen, The Netherlands; rudi.hessel@wur.nl
* Correspondence: abdallah.alaoui@giub.unibe.ch
† These authors contributed equally to this work.

**Abstract:** Recent research established a link between environmental alterations due to agriculture intensification, social damage and the loss of economic growth. Thus, the integration of environmental and social dimensions is key for economic development. In recent years, several frameworks have been proposed to assess the overall sustainability of farms. Nevertheless, the myriad of existing frameworks and the variety of indicators result in difficulties in selecting the most appropriate framework for study site application. This manuscript aims to: (i) understand the criteria to select appropriate frameworks and summarize the range of those being used to assess sustainability; (ii) identify the available frameworks to assess agricultural sustainability; and (iii) analyze the strengths, weaknesses and applicability of each framework. Six frameworks, namely SAFA, RISE, MASC, LADA, SMART and public goods (PG), were identified. Results show that SMART is the framework that considers, in a balanced way, the environmental, sociocultural and economic dimensions of sustainability, whereas others focused on the environmental (RISE), environmental and economic (PG) and sociocultural (SAFA) dimension. However, depending on the scale assessment, sector of application and the sustainability completeness intended, all frameworks are suitable for the assessment. We present a decision tree to help future users understand the best option for their objective.

**Keywords:** agriculture; sustainability frameworks; socio-economic and environmental indicators; soil land management

## 1. Introduction

Agricultural land covers over a third of the earth's surface [1] and 41% of the European Union's 28 member states [2]. Agriculture uses and affects natural resources, such as soil and water, shaping the landscape and contributing to establishing and maintaining semi-natural habitats [3]. Over the last decades, agricultural management practices have changed considerably to enhance crop yields and productivity to ensure food security [4]. This has been achieved through (i) technological developments, particularly by improving and adapting machinery to different management requirements, the genetic improvement of seeds and development of new agrochemicals [5], (ii) the plantation of extensive areas of monocultures [6] and (iii) the high use of mineral fertilizers and phytopharmaceuticals (e.g., pesticides and herbicides) [7–9].

The pressure on the agriculture sector will continue to rise due to global challenges, such as an increasing population and food requirements, and climate change [10]. To

meet the world's projected food demands by 2050, food production must increase by 60–100% [11]. Furthermore, global agricultural production will be affected by increasing competition with certain non-food crops for several economic sectors (e.g., energy for bio-fuels production), a reduction in market prices due to globalization and limited natural resources driven by, e.g., land degradation and water scarcity [12,13] exacerbated by climate change [14].

Agricultural intensification is often associated with environmental degradation, including soil erosion, water, and soil contamination, and biodiversity loss [15–18]. By the end of the 20th century, the consequences of the intensive agriculture approach, especially in developed countries, were thoroughly investigated and frequently reported. As a result, agriculture had been highlighted as one of the main activities worldwide contributing to water depletion [19], soil degradation/pollution [20,21], biodiversity loss [22] and climate change [23]. According to the EU Soil Thematic Strategy [24], the erosion and loss of organic matter are some of the major soil threats affecting agricultural areas, along with compaction, contamination, salinization and loss of soil biodiversity.

Besides environmental problems, intensive agriculture also causes social damage and the loss of economic growth itself in the medium/long term [25]. Thus, the integration of environmental and social dimensions is key for economic development itself, and sustainable agriculture is therefore seen as the only approach towards a successful future [26]. When assessing the sustainability of different agricultural land-uses and land management practices, it is therefore important to consider not only the immediate economic benefit but also how they compromise the overall environmental quality and affect the rural communities, since these factors are relevant to sustaining future economic growth in the short and long-term [27].

As stated in the literature "Sustainability is a multidimensional concept [28] of a dignified life for the present without compromising a dignified life in the future or endangering the natural environment and ecosystem services" [29],. Its evaluation process plays an important role in the development and promotion of sustainable agricultural systems [30]. To investigate the transition towards more sustainable production, various frameworks have been proposed to gain knowledge about the sustainability performance of such production systems [31,32]. Some of these frameworks are based on indicators, whereas others are based on indices (e.g., [33]). Indicator-based sustainability assessment frameworks combining environmental, economic and social issues require the processing of a wide range of information (qualitative vs. quantitative), parameters and uncertainties [34]. They also differ in scope, target audience, indicator selection, aggregation, weighting and scoring methods, as well as the time required to complete the assessment [35]. Although many frameworks emphasize the necessity of including socio-economic and environmental aspects in sustainability assessment, many others focus only on environmental indicators to investigate the short- and long-term effects of different agricultural management practices [36] or are applied to a specific context [37]. In addition, existing assessment methodologies to investigate agricultural sustainability are scattered, focusing on single, complicated and demanding aspects regarding time, cost and required skills.

The main aim of this paper is to identify and summarize the indicators and frameworks used to assess sustainability in agricultural areas. The specific aims are (i) understanding the criteria to select appropriate frameworks and summarize the range of those being used to assess the environmental and the socio-economic themes of agricultural sustainability; (ii) identifying the frameworks available to assess agricultural sustainability; and (iii) understanding the methodological approach and analyzing the strengths, weaknesses and applicability of each framework.

## 2. General Considerations

The following section summarizes the general considerations about the indicators' importance, and selection criteria to set the context for those commonly used in the selected frameworks to assess sustainability in agriculture.

## 2.1. Criteria for Selecting Sustainability Indicators

Indicators are set to monitor and highlight the current conditions and enable stakeholders (e.g., farmers, businesses, policymakers) to identify trends and compare performances among specific places, such as farms, regions or countries, concerning their sustainability performance [38]. They should present the results in a way that is understandable by people with different occupations and sociocultural and educational backgrounds, since they are a powerful public communication tool [39].

The selection of indicators is crucial since it influences conclusions. Thus, the purpose of the assessment, the system boundaries (e.g., aims, scope and temporal and spatial scales) and the end-users should be clearly identified [40]. The assessment should also establish a baseline or reference value (starting point to measure change from a certain state or date) or target (usually established by policymakers). The comparison and contextualization helps to understand the current state or trend [41] and to support the interpretation/significance of the results [39]. Criteria to select indicators include: (i) coverage of environmental, economic and sociocultural dimensions of sustainability [1]; (ii) practicability and simplicity considering field measurements and data availability (e.g., historical data), which should consider spatial and temporal data coverage, reliability, accuracy and consistency [38,42]; (iii) the meaningful use of the indicators to take into consideration the differences in culture and geography to match them to locally relevant problems [39]; (iv) the system's sensitivity to both anthropogenic and natural stresses [1]; (v) meaningfulness to end-users in order to respond to stakeholders' expectations and support policy decisions [40]; and (vi) cost-effectiveness, since the costs to produce the information should justify the benefits of the knowledge produced [40].

Selected indicators can be assessed by qualitative or quantitative techniques [41]. Qualitative techniques are typically based on visual evaluations applied at the field scale and have been increasingly used to evaluate the soil quality (e.g., soil structure and texture, rooting depth and slope) and farm management information [42]. Ball et al. [43] summarized the visual assessment techniques that can be used to monitor soil structure, soil quality and fertility as impacted by land management. Quantitative techniques include: (i) direct measurements via field data collection (e.g., crop yields); (ii) a compilation of secondary data based on a literature review; (iii) statistical correlations of the existing data (e.g., soil compaction); (iv) modeling approaches based on empirical models (e.g., biophysical and economic); or (v) sensing approaches, such as spectroscopic techniques and remote sensing [1].

## 2.2. Indicators Typically Used

Table 1 summarizes chronologically some relevant studies assessing the sustainability of different agricultural practices using indicators. These studies acknowledge the need for a coherent and consistent methodology to successfully evaluate the agricultural management practices and the adoption of three-dimensional indicators. They demonstrate that an oversimplification of the evaluation does not provide a comprehensive overview of the sustainability potential of the different farming practices. These studies also show the myriad of indicators/methodologies that can be used when assessing agriculture sustainability, namely when different farming systems, practices and geographical locations are considered.

Due to the growing concern for environmental issues, numerous indicators devoted to the environmental dimension have been used, and relatively little integration of social and economic aspects on farm assessments has been considered [40]. Environmental indicators reflect the complex interaction between agriculture and environment, providing a cause-and-effect relationship. They tend to include the number and type of crops in the farm, since it links to agricultural biodiversity; soil cover, which is linked to soil erosion; water use; nutrient balance (particularly of nitrogen); and the use of pesticides [44,45], given their toxicity to the environment.

Since soil is nowadays seen as one of the most valuable resources on Earth, given its essential elements to sustain and maintain life, it has received increasing attention under the environmental indicators, and typically includes physical, chemical and biological aspects. Bünemann et al. [46] identified the most frequently proposed soil quality indicators and summarized the measured soil properties that have been used for assessing the environmental dimension in agricultural land uses from 65 soil quality assessment approaches.

Economic indicators aim to address the economic context, focusing on the economic viability defined by profitability, stability, liquidity and productivity, based on input and output prices and yields [47].

Profitability is calculated by cost and revenue, and includes variable and fixed costs (e.g., land rent), whereas liquidity measures the ability of an enterprise/farm to meet short- and long-term obligations and stability is determined by the equity share and equity development [39]. Another important indicator is productivity, which measures the ability of production systems to generate output [48]. Typical economic indicators also consider public subsidies for the farmers, since they provide protection regarding their agricultural activities. GDP is sometimes considered as an indicator of the difference between producers' income and transfers to other economy sectors (variable costs, subsidies) [44].

Most social indicators focus on the following: (i) the sustainability of the farming community, which involves the welfare of the relevant actors and communities; and (ii) the sustainability of society as a whole. The first type of indicators focuses mainly on working conditions, education and the quality of life defined by physical well-being and psychological well-being [40]. Social sustainability is linked to society's demands, with regards to its values and concerns [49], and may be grouped in: multifunctionality (e.g., quality of rural life, contribution to local employment and to ecosystem services) [50], sustainable agricultural practices (e.g., animal welfare, environmental impacts) and product quality (e.g., quality processes, food safety) [40]. These indicators also tend to measure the socio-economic implications of agriculture in the rural income, and may be measured by the total labor generated, as well as by the seasonal variations linked to individual crop requirements, often associated with peaks in agricultural employment (e.g., sowing and harvesting) [44]. Measuring indicators of a sociocultural dimension is challenging, since they are based on a qualitative assessment and are therefore subjective. Farm-community-based indicators are usually based on farmers' self-evaluation gained from surveys or interviews [40].

**Table 1.** Most relevant papers on sustainability providing relevant information on the indicators and their relevance for case study applications and different conditions. The papers in relation to the frameworks considered in this manuscript are not reported, except those comparing different frameworks.

| Reference | Summary of the Study |
|---|---|
| [50] | Presents the farmer sustainability index (FSI), relying on sustainability scores for diverse agricultural management practices to avoid an oversimplification of the reality. The study focuses on 33 production practices implemented by [51] Malaysian farmers to assess the FSI scores. |
| [39] | The sustainability of the agricultural systems is assessed based on different points and levels, considering the need to improve the assessment methods used for some agricultural sustainability subthemes. The limited availability of tools to evaluate qualitative aspects, such as landscapes and animal welfare, was identified as a major shortcoming. It also highlights the need to couple economics and social sciences with environmental processes for a better understanding of the overall agricultural system. |
| [52] | By analyzing the impact of agri-environmental indicators (AEIs) on policy outcomes, the paper examines the potential impacts of Agri-environmental Regulation EC 2078/92 on European agricultural landscapes. It discusses the frameworks divided in policy outcomes and policy performances and analyzes the obstacles to measuring policy outcomes directly. The study focuses on intensification and abandonment problems in extensive agricultural areas of Spain and Denmark. |

**Table 1.** *Cont.*

| Reference | Summary of the Study |
|---|---|
| [53] | The environmental impacts of agriculture are investigated through life cycle assessment (LCA). The LCA framework was adapted in terms of functional unit and impact categories of the agricultural production process. The framework was applied in 18 grassland dairy farms managed under different intensity levels in southern Germany. |
| [54] | Investigates a method for evaluating the environmental impacts of arable farming systems. The method is based on agro-ecological indicators (AEI) to rank or classify the cropping systems. The agro-ecological indicators tested include phosphorus and nitrogen fertilization, irrigation, pesticides, organic matter, cropping pattern, crop succession and covering, ecological structures, soil management and energy. |
| [55] | Environmental impacts, economic viability and social acceptability are investigated in two production systems. The sustainability of the system is based on 12 indicators assessed through empirical data from household survey, soil samples, field observations and information supplied by key informants. Management of soil fertility, pests and diseases, the use of agro-chemicals and crop diversification were significantly different between both systems. In turn, indicators, including crop yield and stability, land-use pattern, food security and risk and uncertainties, showed similar results. |
| [56] | The use of pesticides, nutrients and energy in 55 farming systems was compared using input–output accounting systems (IOA) covering the topics of the farm's use of nutrients, pesticides and energy. The indicators and approach used varies from systems using physical input–output units to systems based on good agricultural practices (GAP). |
| [57] | Proposes the Sustainability Assessment of Farming and the Environment (SAFE) framework, aiming to assess the sustainability of agricultural systems through several criteria and indicators. The framework can be applied at different spatial scales, including parcel, farm, landscape, region or state. This is a hierarchical framework, comprising structured principles, criteria and indicators. SAFA serves as an assessment tool for identifying, developing and evaluating the overall sustainability of agricultural systems, techniques and policies. |
| [58] | It presents the Indicateurs de Durabilité des Exploitations Agricoles or Farm Sustainability Indicators method (IDEA) tool, which includes 41 sustainability indicators, and is devoted to supporting farmers and policy makers. The study reveals that the IDEA method requires adaptation of indicators to local farming. |
| [48] | Based on an irrigated agriculture area in Spain, authors perform a comparative analysis of different methods for developing composite indicators to analyze agricultural sustainability. The study uses indicators calculated from several farms and policy scenarios. |
| [59] | Develops a methodology to evaluate the sustainability of two agricultural systems in Spain (rain-fed vs. irrigated) through composite indicators. It reveals farm heterogeneity in each individual agricultural system in terms of sustainability, and analyzes the influencing variables to support decision making. |
| [60] | Proposes a framework for an integrated assessment of sustainability in European regions and policy options. The framework is used in ex ante assessment of land use policy scenarios and includes environmental, economic and social aspects in different sectors (forestry, agriculture, tourism, transport and energy). The conceptual framework can be applied at different scales (regional, European), and considers the variability of the European regions. |
| [61] | Presents a project funded by the UK government to develop a methodology for assessing the sustainability of both conventional and organic farming systems. The project includes 40 environmental, social and economic indicators. Data were collected to support the chosen indicators. The selected set of indicators assesses the advantages and disadvantages of the different farming systems, and the results can be useful to improve the sustainability of the farming systems. |
| [62] | Provides a review of current management tools to address sustainability in small and medium-sized enterprises (SMEs) and highlight the advantages of such tools for SMEs. Results show that most tools are not implemented by the majority of SMEs, and summarize the barriers for this. The paper also suggests criteria to facilitate future implementation. |
| [63] | The MASC framework is used to evaluate the performance of 31 agriculture cropping systems. Conservation agriculture displayed a greater sustainability performance, especially regarding the environmental criteria. However, conservation agriculture systems revealed several weaknesses, namely regarding those of technical or social nature. |

**Table 1.** *Cont.*

| Reference | Summary of the Study |
|---|---|
| [64] | Four sustainability assessment tools (RISE, SAFA, PG and IDEA) were compared regarding the indicators used for perceiving practical requirements, procedures and the complexity of their application on five Danish farms. The scoring and aggregation method used in each tool vary widely, as well as the data input and time requirements. RISE was considered as the most relevant tool. However, farmers seem hesitant in applying the outcomes of the tools to support decision making and management. |
| [65] | Develops a set of indicators based on generally available data to assess the sustainability of urban food systems. Through a participatory process, an assessment method considering 97 indicators for evaluating 51 of the 58 subthemes was considered developed. The method was tested in Basel city, Switzerland, and revealed that it was useful to improve the sustainability of the tested investigated food system. |
| [66] | By using a set of environmental, social and economic indicators, the sustainability of an agricultural sites in Italy was assessed. The indicators were identified based on IDEA, RISE, SAFE, SOSTARE and MOTIFS methodologies. The framework developed provides easy-to-read results relevant for different scales assessment, and relies on balanced features of data availability and reliability. |
| [67] | The environmental sustainability of the ornamental plant production sector (including both nurseries growing plants in container production (CP) and in open field (FP)) is assessed through impact indicators. The results exposed the higher environmental impacts of the CP comparing with the FP due to their peculiar production structure, which, thus, must be improved to assure an acceptable environmental performance. |
| [68] | The social sustainability of the Swedish (livestock) farming system is investigated using the social indicators considered in existent sustainability assessment tools (RISE, SAFA, IDEA). From these three tools, RISE seems best at capturing the social situation of the farmers, although not fully addressing the finding work aspect. Both SAFA and IDEA fail to capture several aspects relevant to describing the situation of the farmers. |
| [69] | Investigates how existent sustainability assessment tools support decision making regarding management practices by farmers. It shows that farmers need more basic and rapid overviews of the complexity dimension, whereas the management dimension is useful to develop and implement new farm strategies. |
| [70] | An ex ante evaluation of several conventional practices is used to enhance the sustainability of cropping systems. The sustainability of five diversified cropping systems is compared with less diversified systems in several arable areas of France. The diversified systems revealed fewer greenhouse gas emissions, improved water and air quality and a high biodiversity. Nevertheless, diversification can cause negative impacts in some indicators, such as $NH_3$ volatilization, $NO_3^-$ lixiviation, pesticide use and gross margin. |
| [71] | A multi-criteria analysis (MCA) tool is developed to assess the sustainability of four Italian organic farms with durum-wheat-based crop rotations. The best sustainability scores were noticed in both ex ante and ex post analysis by diversified cereal farming systems with short supply chain mechanisms to sell their products. |
| [72] | A sustainability assessment of the flowering potted plants (FPP) value chain was performed, including all of the phases from breeding to distribution. The selected indicators relied on SAFA and RISE sustainability assessment tools. The study shows that SAFA and RISE tools do not cover the overall sustainability subthemes, and emphasizes the need for a system-specific view in unique systems, such as the FPP. |
| [73] | The relationship between agricultural sustainability and economic resilience is investigated through an empirical analysis of Northern European countries. Composite indicators are settled based on decision-making criteria. Results highlight that sustainability indicators cannot be replaced by economic resilience ones, and that the latter should be considered in addition to the economic sustainability indicators. |

## 3. Methodology

During the past 20 years, various approaches and tools have been proposed for assessing the overall sustainability in the agricultural production system and food sector [31,74,75]. However, these methods have many limitations. As an example, life cycle assessment tools quantify many aspects of the environmental dimension in a narrow way, need a high amount of data and do not consider the impacts on soil quality and biodiversity [76] and economic and socio-cultural impacts [77], or can only be applied to agricultural enterprises [32]. Eco-management and audit schemes, as well as sustainability reporting systems, include procedures accounting for the sustainability of a company, but do not

enable comparison between the outcomes of different ones since they are not science-based assessments [78].

In this study, we selected indicators and frameworks based on the following criteria: (1) went through a peer review process, (2) have a farm assessment level, (3) cover universal agricultural sectors, (4) include the three dimensions of sustainability, (5) suitable for Europe and countries worldwide and (6) present transparency of information allowing for an informed assessment as well as solid cultural and value-based elements.

For the search of the frameworks, we considered literature including at least one peer reviewed publication, reports and presentations available online by searching on scientific web platforms.

Each framework selected was therefore described by stating information on the type of tool used (software, database, etc.) and where it can be found available, requisites for running the tool, type of input data required, time needed for the assessment and number and description of indicators (environmental, socio-cultural and economic) used.

The six sustainability assessment frameworks were also compared according to their ability to cover the main themes of environmental, economic and sociocultural dimensions, and their themes were reported. We compared their strengths and weaknesses and developed a decision tree based on possible scales, sectors of applicability and the completeness of sustainability dimensions required to help stakeholders decide which framework is the most suitable for their sustainable assessment purposes.

## 4. Results

Based on selected criteria, the following frameworks were identified: SAFA, RISE, MASC, LADA, SMART and PG. Below, each framework is briefly described, as are the environmental (Table 2), sociocultural (Table 3) and economic (Table 4) indicators included in each one of them. In the next section, their strengths and weaknesses are highlighted individually.

**Table 2.** Environmental themes, sustainability objectives, indicators and measured parameters for each framework considered in this study.

| Theme | Sustainability Objectives | Indicators | Framework | Parameters |
|---|---|---|---|---|
| Water use | Water conservation | Water management | RISE | Water consumption monitoring and measures for water saving |
| | | | PG | Irrigation, flooding defences, pollution reduction, water management plan |
| | | | SAFA | Reduction in water consumption/water withdrawals |
| | | Dependency of water | MASC | Irrigation, water availability and crop water requirements |
| | Water security (supply without compromising available water resources) | Water Supply | RISE | Assessment at watershed scale |
| | Availability of water resources for irrigation, salinization | Irrigated areas | LADA | Water availability |
| Water quality | Water resources degradation | Overexploitation of water resources, salinization | LADA | Groundwater level, salinity of water, arsenic contamination |
| | | Clean water target | SAFA | Concentration of water pollutants, wastewater quality |
| Water pollution | Water pollution risks | Pesticides losses in water | SMART | $NO_3$ losses, phosphorus losses |

**Table 2.** *Cont.*

| Theme | Sustainability Objectives | Indicators | Framework | Parameters |
|---|---|---|---|---|
| Soil quality/land degradation | Providing the best conditions for plant growth and soil health, preventing land degradation | Physical and chemical properties | SMART | Compaction, erosion, SOC, phosphorus fertility, |
| | | | PG | Cultivation, winter grazing, NPK management, cropland diversity, livestock diversity |
| | | | RISE | Soil reaction |
| | | | SAFA | Soil chemical and biological quality, soil structure and SOM |
| | Identification of soil and terrain resource degradation | Erosion, compaction, nutrient and soil biodiversity decline, salinization (regional) | LADA | Texture, structure, pH, organic matter, water infiltration/drainage, salinity, soil depth, landslides, gullies |
| | | | RISE | Soil erosion, soil compaction |
| | | | SAFA | Soil health, soil degradation, net loss/gain of productive land |
| | | Soil resources (local) | LADA | Heavy metals, earthworms (and others), root development, soil color |
| Air quality | Prevention of air pollutant emissions and elimination of ozone-depleting substances | GHG, air quality | SMART RISE | Air pollution, ozone substances, GHG |
| | | | SAFA | Emission of air pollutants, number of days of the year with exceedance of air pollution values, GHG emission, net direct GHG emission |
| Climate | *Climate resources:* Identification drought/desertification and water erosion | Aridity, soil moisture, variability of rainfall | LADA | Aridity index, soil moisture change, inter-annual and trends of rainfall |
| | Extreme events: Tsunami, heavy rains, long drought, dust storms, volcanic eruption, water erosion | Extreme events, disasters, slope/land use | LADA | Salinization, landslides, loss of land cover and biodiversity, sedimentation |
| Plant and fertility | Fertilizer conservation: Prevent nutrient losses through runoff | Wastewater quality | SAFA | Nitrate and orthophosphate concentrations |
| | | | RISE | Material flows, fertilisation Environment pollution |
| | Abiotic resources conservation | Phosphorus conservation | SMART | Crop phosphorus needs, phosphorus use autonomy |
| | | | PG | Manure management |
| | *Reduce plant protection:* Reduce application of chemicals and avoid environmental exposure | Plant protection Practices [1] | RISE | Agreement with integrated plant protection principles |

| Theme | Sustainability Objectives | Indicators | Framework | Parameters |
|---|---|---|---|---|
| Biodiversity | Preserve diversity of ecosystem, species and generic | Species conservation practices | SMART | Conservation of functional integrity, agrifood ecosystem, wild and domesticated species |
| | | | PG | Conservation plan, habitats, rare species |
| | | | SAFA | Rare and endemic species, wild animals, threatened or vulnerable wild species |
| | | Functioning and connectivity of ecosystem services | SAFA | Ecosystem services, connectivity, structural diversity of ecosystems, land-use and land-cover change |
| | Preserve vegetation resources | Changes in land cover | LADA | Loss of biodiversity/loss of nutrient |
| | | Genetic diversity | SAFA | Wild genetic diversity, agro-biodiversity, locally adapted varieties/breeds, rare and traditional varieties and breeds |
| | Pesticide use intensity | Number of doses | MASC | Sprayed area, insecticides, fungicides, herbicides |
| | Infrastructure and production | Management and production | RISE | Management of biodiversity, ecological infrastructure, distribution of ecological infrastructures, diversity and intensity of agricultural production |
| Energy use (temperature control/heating storage and transport) | Reduce GHG emissions and energy consumption | Measures to save energy | SAFA | Implementation of energy-saving practices |
| | | | PG | GHG emissions |
| | | Energy conservation | MASC | Energy consumption, energetic efficiency |
| | | | PG | Energy balance, benchmarking |
| | | | RISE | Energy management, energy intensity, greenhouse gas balance |
| | Reduce non-renewable energy sources' dependency | Renewable energy | SAFA | Net of energy use and share of sustainable energy transports |
| | Waste reduction and disposal | | SMART | Prevention of waste generation |
| | | | PG | Disposal of farm waste |
| Energy use Substrate and containers | Reduce non- renewable materials (e.g., plastic, peat) | Material consumption practices | SAFA | Replacement of non-renewable materials by renewable and recycled materials |
| | Reduce non-degradable waste such as plastic or substrate (perlite) | Waste reduction practices | SAFA | Reducing the generation and hazardousness of waste, food loss and waste reduction |

**Table 2.** *Cont.*

| Theme | Sustainability Objectives | Indicators | Framework | Parameters |
|---|---|---|---|---|
| Animal welfare | Animal health and freedom from stress | Animal health | SMART | No thirst, hunger, injury and disease |
| | | | PG | Housing, bio security, ability to perform natural behaviors |
| | | | RISE | Animal production management, productivity of animal production, possibility of species-appropriate behavior, living conditions, animal health |
| | | | SAFA | Reduce pain and injury risk of animals, condition of animal husbandry |

[1] Originally "Plant protection" in the RISE framework.

**Table 3.** Social themes, sustainability objectives, indicators and measured parameters for each framework considered in this study.

| Theme | Sustainability Objectives | Indicators | Framework | Parameters |
|---|---|---|---|---|
| Employment contract/agreement | Workers' stability and secure workplace through legal contracts | Employment relations; ability to cover the costs of production, right of suppliers | SAFA | Written agreements with employees |
| | | No forced labor, no child labor, freedom of association and right to bargaining | SMART | Fair prices, rights of suppliers are respected, labor rights |
| Workload | Allows overtime compensation and quality of life | Working hours | RISE | Working hours and vacations recorded and following the standards |
| Wages | Wages provide reasonable life quality for workers and their families | Wage level | SAFA SMART | Living wage paid to employees |
| | | Profession and education, financial situation, social relations, personal freedom and values, health | RISE | Education, economic and social situation, health |
| Health safety | *Occupational health and operational difficulties:* Employees trained for health and safety issues/complexity of implementation | Safety and health trainings/health risks | SAFA MASC | Existence and effectiveness of employees' health and safety training/physical constraints, number of specific operations, number of crops |
| | Safe working environment | Safety of workplace | SAFA SMART | Determining safe, clean and healthy workplace |
| | *Medical care:* Access to affordable medical care for employees; | Health coverage and access to medical care | SAFA | Employees' access to medical care; and health provisions |
| Job satisfaction | Attract and retain employees | Capacity development | SAFA | Opportunities for employees' capacity development and advancement |
| | | | PG | Skills and knowledge |
| Decent livelihood | Enjoy a livelihood, time for culture and nutritionally adequate diet, training and education, access to means of production | Life quality, development capacity, fair access to production income | SMART | Adequate livelihood, possibilities for education and training, access to production means |

**Table 3.** *Cont.*

| Theme | Sustainability Objectives | Indicators | Framework | Parameters |
|---|---|---|---|---|
| Gender equality/equity | No gender discrimination, including support of working mothers through provision of maternity leave; non discrimination, support to vulnerable people | Gender equality equity, non-discrimination | SAFA SMART | Resources to provide women's pregnancy rights; equity and non-discrimination policies are taken into account; disadvantaged groups are promoted and supported. |
| Cultural diversity | Freedom of choice and ownership in regards to production means | Indigenous knowledge, food sovereignty | SMART | Intellectual property right, choice and ownership in regards to production means |
| Benefits to/investment in local communities | Support of/invest in local communities | Community investment | SAFA | Investment to meet local community needs |
| Employment | Contribution to local/ regional employment | Regional workforce | SAFA MASC | History of preferential hiring of local employees when possible, |
| | | | PG | Community engagement |
| Consumer safety | Product free of highly hazardous pesticides | Hazardous pesticides | SAFA | Any highly hazardous and other pesticides used (safety to consumers and pollinators) |
| Transparency | Consumer informed of product quality through a reliable labeling system | Product labeling | SAFA | Products are labeled in compliance with standards |

**Table 4.** Economic themes, sustainability objectives, indicators and measured parameters for each framework considered in his study.

| Theme | Subtheme | Sustainability Objectives | Indicators | Framework | Parameters |
|---|---|---|---|---|---|
| Profitability | Net income/autonomy | Maintain short- and long-term profitability of the business/autonomy | Net income | SAFA MASC SMART | Total revenue in the last five years associated with producing goods and services exceeds the totalprofitability, independency, efficiency, specific equipment needs |
| | | | Liquidity, stability, profitability, indebtedness, livelihood | RISE | Liquidity, stability, indebtedness, livelihood |
| | Profitability per unit product | Costs of unit production are lower than the price per unit of product sold | Cost of production | SAFA RISE | Cost of the products sold per unit of production, break-even point |
| | | | | PG | Financial viability |
| Vulnerability | Stable production | Mitigating production risk such as unpredictable weather conditions and pathogen infestation | Production risk [1] | SAFA | Implementation of mechanisms to prevent disruption of volume or quality |
| | | | | SMART | Stable business relationships and accessibility to alternative procurement channels |
| | | | | SAFA | Procurement channels to reduce the risk of having input supply shortages, stability of supplier relationships |
| | Assortment | Diversified products to ensure market growth, product differentiation and reduced risk (market, weather, price) | Product diversification | SAFA | Number and type of products, as well as development of new products |
| | Diversified income | Diversified income structure (marketing channels and buyers) and production contract with buyers | Stability of market | SAFA SMART | Activities to diversify marketing channels and stabilize prices |
| | Risk management | Internal and external risks (e.g., demand uncertainty, shortage in workforce) | Risk management | SAFA SMART RISE | Existence of a plan or a strategy to reduce risks and adapt [3] |
| | | | | PG | Farm resilience |
| | Liquidity | Financial liquidity to withstand shocks | Financial liquidity [2] /independence | RISE MASC SMART | Cash flow plus available credit lines divided by average weekly expenditure |
| | | | | SAFA | Net cash flow, safety nets |

**Table 4.** *Cont.*

| Theme | Subtheme | Sustainability Objectives | Indicators | Framework | Parameters |
|---|---|---|---|---|---|
| Accountability | Product traceability, food safety and quality | Products can be traced along the value chain | Traceability system | SMART | Share of production that can be traced along the value chain, food safety and quality |
| | | | | SAFA | Product labeling, traceability system, certified production, food quality, control measures, hazardous pesticides, food contamination |
| | | | | PG | Food quality certification |
| Investment | Internal, community, long-ranging investment | Sustainable performance and development of a community aiming at long-term sustainability | Resilience | SMART | Enhancing sustainability performance, sustainable development of a community, long-term sustainability |
| | | | | SAFA | Long-term profitability, business plan |
| | | | Internal investment | SAFA | Improved social, economic, environmental and governance performance |
| | | | Community Investment | SAFA | Balance between the community needs and efficient use of environmental resource |
| Local economy | Value creation, local procurement | Benefit of the local economies through procurement from local suppliers | Local economy | SMART | Benefit to local economies through employment and payment of local taxes, |
| | | | | PG | Local food, production of fresh produce |
| | | | | SAFA | Regional workforce, fiscal commitment, local procurement |
| Economic risk | Loss of land | Identification of the risk related to the loss of profit | Frequency of forest fires, presence of land mines, under-management resource, urbanization, livestock pressure, human-induced disasters | LADA | Deforestation, complete loss of land, nutrient loss/erosion, sealing, compaction, loss of land cover, isotope fall out (radio nuclear) |
| | | | | PG | Landscape features, management of boundaries |

[1] Originally "guaranty of production levels" in the SAFA framework. [2] Originally "liquidity" in the RISE framework. [3] Addressed by operational management with the indicators: goals, strategy and implementation, information availability, risk management and sustainable relationships. SMART has a 4th dimension "Good Governance", with the following themes: corporate ethics, accountability, participation, rule of law and holistic management (not included here).

### 4.1. SAFA

The Sustainability Assessment of Food and Agriculture systems (SAFA) is a framework developed and proposed by FAO to assess the environmental and social impacts of food and agricultural operations [79]. It offers a comprehensive reference framework for assessing sustainability in agricultural, forestry and fishery chain systems. The framework is designed hierarchically starting with four dimensions: environmental integrity, social well-being, economic resilience and good governance [72].

The available software (https://www.fao.org/nr/sustainability/sustainability-assessments-safa/safa-tool/en/) (accessed on 4 March 2022) calculates 116 indicators that target the principles of sustainable development. Measured and/or calculated data from production sites with defined unit processes of a system include a wide diversity of sources, including literature or available databases, and public and other independent

sources of information. Additionally, interviews are carried out with local employees in the sector considered. Data analyses should be conducted by an expert in sustainability. SAFA-Tool assists users with setting system boundaries and scoring ranges, and selecting targets, practices or performance indicators from qualitative or quantitative information. The latest software version 2.4.1 allows the user to add their own indicators. Depending on the complexity level of the analysis, determined by the choices made by the user, data collection may range from ±2 h to weeks, and the total assessment from 0.5 days to months [69].

Environmental indicators established in SAFA cover a broad range of themes including water use, wastewater quality, soil quality, air quality, species conservation practices and ecosystem diversity, energy-saving practices, material consumption and reduction practices, energy use and animal welfare, all linked to the food and agriculture processes (Table 2). The social angle of the evaluation process is also very well represented in SAFA, with the rating of indicators covering themes such as employment contracts, the wage level of employees, safety and health environment, job satisfaction, gender equality, cultural diversity or even transparency in the labeling, safety for the consumer and the impact of using a regional workforce (Table 3).

Finally, economic indicators figuring in SAFA cover both profitability and vulnerability topics, such as the net income, production cost and risk and stability of the market or risk management, among others. It also includes indicators related to accountability, such as the existence of system traceability, the investment potential and the will to invest in local economy (Table 4).

LADA data are extracted from the LADA indicators' toolbox developed for LADA (see [80]); the indicators of LADA are divided into two types: those describing the state of the resources+ and those describing direct pressure on the resources++; thus, the indicators used are those that indicate the degradation type

### 4.2. RISE

The framework RISE (Response-Inducing Sustainability Evaluation) was developed by Hafel, in Switzerland, for evaluating the environmental, sociocultural and economic sustainability of farm operations [80]. Currently, the RISE version 3.0 software can be found online (RISE 3.0 - Software Manual (bfh.ch)) (accessed on 4 March 2022) or offline (Microsoft SilverlightTM plug-in required) to analyze the data. It includes a total of 50 indicators addressing environmental, social, economic and land management aspects. The data are collected with a questionnaire-based methodology, where farmers are interviewed for 3 to 5 h, which, with the additional time for data computation, requires a total assessment time of 5–9 h [80]. The framework should be used by agronomists or specialists in agricultural advisory. The results are thoroughly discussed with farmers and used to support the continuing improvement of farm sustainability. The environmental indicators included are mainly related to water use and plant protection (Table 2), whereas the social dimension is focused on the workload and the economic dimension mainly tackles the business vulnerability by assessing the financial liquidity (Table 4).

### 4.3. MASC

INRA (Institut National de la Recherche Agronomique) developed MASC (Multi-attribute Assessment of Sustainability of Cropping Systems) to assess how cropping systems contribute to sustainability at the farm level [13]. The tool that is currently available (http://wiki.inra.fr/wiki/deximasc/Main/) (accessed on 4 March 2022) uses a decision tree to break down the sustainability assessment decisional issue into 32 input criteria. Indicators used to assess these basic input criteria can be chosen by the user depending on their accuracy and the context of their study, as well as the available data [63].

Qualitative and quantitative information is collected through questionnaires and reported results. Methods such as MASC that are suited for the analysis of qualitative data may be more relevant for sorting and categorizing technical solutions when con-

sidering a wide range of performances [13,81]. The tool should be managed by a researcher/professional, who then interprets the results obtained.

The indicators included in this framework deal with the evaluation of environmental aspects such as water use, biodiversity and energy use through indicators of water dependency, number of pesticides doses and energy conservation (Table 2). Social indicators are also included, especially targeting the safety and health trainings of employees and the priority to employ a regional workforce. The economic dimension is assessed through indicators of net income and financial liquidity (Table 4).

### 4.4. LADA

The LADA tool (Land Degradation Assessment in Drylands) framework was developed by FAO (Food and Agriculture Organization of the United Nations) for assessing and quantifying the nature, severity, impact and extent of land degradation on ecosystem services across different spatial and temporal scales. In order to support policy decisions to combat land degradation, the framework aims to identify hotspots and bright spots [82]. It is available as a tool-kit (https://www.fao.org/nr/kagera/tools-and-methods/lada-local-level-assessment-manuals/en/) (accessed on 4 March 2022) that identifies the state of the land resources through different indicators, the pressures and driving forces that caused this status and the impacts on ecosystem services and on livelihoods. The data required are collected through agricultural and other national surveys and censuses and maps of soil and natural resources, as well as digital and computer-assisted methods.

LADA environmental indicators focus on water quality and water use, soil quality and the soil degradation status. It includes an assessment of the irrigation area and the over-exploitation of water resources, as well as the salinization process, and includes indicators focused on general soil threats, including erosion, compaction and nutrient loss. Biodiversity is also tackled through indicators of land cover (Table 2). Additionally, LADA also includes economic indicators related to the economic risk caused by land degradation problems, through the assessment of land loss by fires, urbanization and livestock pressure, among others (Table 4). The sociocultural dimension is represented by the pressures on the resources that will impact society as a whole. The change in land users' life is not investigated. The LADA framework considers climate components illustrated by climate resources and climate extreme events.

### 4.5. SMART

The SMART (Sustainability Monitoring and Assessment RouTine) framework was developed by FiBL (Research Institute of Organic Agriculture) to assist farms and enterprises in the food sector for assessing their sustainability level in a credible and transparent manner [83]. The specific software (https://www.fibl.org/en/themes/smart-en/smart-method) (accessed on 4 March 2022) is used to compute context-specific indicators (up to 200) that are compiled individually for each case study. Data needed for the assessment are semi-quantitative and collected using a standardized interview procedure [84]. The time for data collection is 2–3 h [64]. The software should be handled by scientists and/or field practitioners. The extensive list of indicators includes transversal environmental topics from water pollution to soil quality and degradation, air quality, fertilizer consumption, biodiversity, energy use and even animal welfare. Examples of the broad list of environmental indicators in the framework include pesticide presence in water, greenhouse gas emissions, phosphorus crops content, conservation of species and the use of renewable energy (Table 2). Social indicators are also included in the framework, assessing employees' rights and their wage level for a dignified life. The social dimension also includes gender equality and non-discrimination, cultural diversity, health coverage and access to medical care (. Finally, economic indicators cover a set of themes, from profitability to vulnerability, accountability, the resilience of the investment and the value of local economy (Table 4).

*4.6. PG*

PG (public goods) is a framework developed by the Organic Research Centre in the United Kingdom for assessing the provision of a broad range of public goods from farming activities [84]. It is based on the premise that agriculture produces many by-products that are deemed public goods [85].

Information related to the farming activity is gathered and computed in an excel sheet (https://www.organicresearchcentre.com/our-research/research-project-library/public-goods-tool/) (accessed on 4 March 2022), where 11 individual public goods are scored. Information is collected using questionnaires with several key "activities" and includes qualitative and quantitative data. The analysis is normally undertaken by famers and/or sustainability experts. The time of data collection varies between 2 and 4 h [84].

Environmental indicators from PG framework include water management and soil quality through the assessment of the irrigation method used, flooding defenses implemented and the existence of water and nutrients management plans, cultivation types and cropland and livestock diversity. Biodiversity and energy use are also tackled extensively through the screening of conservation plans, the presence of habitats and rare species, GHG emissions, energy balance and the correct disposal of farm waste. The animal welfare is accounted through parameters such as housing, biosecurity and their ability to behave naturally (Table 2). The social indicators are basically represented in the job satisfaction through the skills and knowledge of the employees and the contribution to local/regional employment assessed by the level of community engagement. Economic indicators range from financial viability and farm resilience to others, such as accountability by food quality certification, the local economy value through assessing the production of local products and the economic risk by checking landscape features and the management of boundaries (Table 4).

## 5. Discussion
*5.1. Strengths and Weaknesses of the Frameworks*
5.1.1. SAFA

The study by Landert et al. [83] aimed to transform intensive livestock farming in 15 European countries with a high impact on the environment, society and economy in sustainable livestock farming, which reduces emissions and the costs associated with this. The authors showed that farms with an optimized governance component can improve sustainability in general and that the farmers should learn about this context and improve their production and economic performance within each individual farm. In this context, SAFA is an important tool to provide recommendations for future actions to support achieving sustainability [86].

The study by [87] in the central Sicily Mountains showed that a growing economy would also require more resources to reduce environmental impacts, modernize animal shelters and use renewable energy sources to make them more sustainable. It illustrates how, on the one hand, the sustainability areas that are discussed in SAFA are interconnected, and, on the other, that there are many open pathways for Sicilian organic farms to improve their performance. Although SAFA is a valid asset for addressing the sustainability potential of food in urban system contexts, two main weaknesses related to some subthemes have been pointed out by Landert et al. [83]: (i) the subtheme Remedy, Restoration and Prevention would need a specific adaptation to become food-focused, and (ii) the subtheme Rights of Suppliers does not include the full web of existing relations and processes normally present in these systems. In addition, the subthemes Long-Ranging Investment, Profitability, Stability of Supply, Stability of Market and Liquidity are not flexible for use in this system [65]. In this context, by setting the boundaries of the system, the majority of the indicators became less responsive to drivers or pressures. In turn, this led to poorer analyses of the cost-effectiveness and political and societal acceptance.

### 5.1.2. RISE

Grenz et al. [88] showed that RISE is an effective tool for field production since it measures fertilizer application relative to soil nutrients and crop requirements for optimum crop growth and calculates the non-renewable energy percentage, as well as the farm financial security (e.g., diversifying income sources, securing access to land, maintaince of infrastructure). Röös et al. [68] observed the potential of this framework to integrate the social dimension of the farm, although some modifications would be necessary to enhance its relevance for the specific context of the study. The authors perceived the results of RISE as highly solid because they are based on quantitative data input and integrate experts on the subject.

RISE becomes complex due to complicated calculations and the elevated number of data required. However, regarding the tool, farmers consider it as relatively simple to understand [68] because of the language adopted compared to the more general one found in in SAFA (e.g., rule of law) [88] and IDEA (e.g., organization of space) [68]. Regarding the relevance of RISE, the farmers recognize that the obtained outcomes reflect the positive and negative points of their farming activities well. Therefore, in comparison to other frameworks (e.g., IDEA, PG), farmers consider RISE as one of the most appropriate frameworks to use [72]. However, it was shown that the time investment and time required for learning RISE are relatively long in comparison to other frameworks [68], while also not being highly transparent as other frameworks due to the complexity of the calculations that complicates the computation rationale behind it [64]. In addition, using standardized quantitative measures makes it hard to capture the specific situation (e.g., farmers' financial situation and working situation), since farming activities will always endorse high variability from one case study to another [68].

Havardi-Burger et al. [72] showed that the process of selecting indicators in RISE becomes difficult since, on the one hand, one must include all of the significant indicators that represent the system well, but, on the other, the number of indicators cannot be too high otherwise it compromises the application of the tool. This aspect is observed for all frameworks except for SAFA, which includes a relatively high number of indicators. In describing this difficulty, Binder et al. [31] refer to parsimony as a principle in order to strive for the system representation under consideration and the sufficiency to address its complexity. Overcoming this difficulty by setting different indicators from different sustainable dimensions and themes is not an easy task since one becomes easily lost on what is actually under study. One possible example is the indicator stability used in RISE to address how financially stable a farm is (e.g., farm infrastructure, long-term access to land, the number of customers and main source of income). The authors showed that covering more aspects would be a benefit, as also shown in the indicator liquidity combining two SAFA indicators (safety nets and net cash flow). This allows the adoption of concrete measures to improve the business performance, even when under financial stress [31].

### 5.1.3. MASC

MASC can be described as an objective and broad tool. Its ability to incorporate qualitative data in addition to its ease-of-use in terms of the necessary input becomes very helpful for real situations and enables a high comprehensibility of the outputs. Quantitative values can be processed as qualitative information by simply using thresholds, and, thus, MASC integrates both measurements (e.g., yields), calculated data (e.g., semi-net margin) and empirical knowledge (e.g., physical difficulties of crop interventions) into the indicators. This ensures that the best available information is used and that there is a high participation approach, since, as an example, the users' point of view can be integrated in the framework, since normally it would be difficult to address them by using quantitative indicators [63].

Graheix et al. [62] applied MASC to evaluate 31 cropping systems previously chosen to study different management practices, from conventional tillage systems to other systems where conservation agriculture principles were incorporated. In this study, the integrative approach of the MASC framework provided a benefit for the understanding of how the

different cropping systems behave when considering, at the same time, (i) the multiple objectives of the dimensions (economic, social and environmental); (ii) various time scales and (iii) the objective worries and goals of the farmers, and generally also the society, raised by different stakeholder groups with various interests. While the results of many studies have highlighted advantages of MASC for adapting cropping systems through conservation agriculture [63], they also identify a weakness in terms of MASCs' inability to properly evaluate the agronomic effects of biodiversity (e.g., normally, a higher biodiversity is an advantage, but decreasing the soil tillage may also contribute to a higher diversity of pests and weeds) from a simple description of the practices employed. The diversification systems may have many advantages (e.g., lower GHG emissions) in comparison with the conventional reference system. They may improve both the air and water quality and contribute to a higher biodiversity [70]. The indicators were initially determined based on scientific knowledge and the context available at the time of the development of MASC, with the aim of keeping its use relatively simple [25]. This probably led to a too generalized meaning of the indicators that cannot highlight the specific context found in different pedoclimatic conditions and under different agricultural management practices of the different studies. As reported by Médière et al. [89], "we still have little scientific information concerning the responses of biological process to agricultural practices in a given pedoclimatic context". The balance between benefits from the services provided and the negative effects that are often observed when tillage is reduced is still unknown, and crop rotation is included, which results in a higher biodiversity [90]. Al Shamsi et al. [91] showed that the best practice reduces the need for off-farm inputs while increasing the product range. However, it is also reported that this diversification can cause negative impacts, i.e., $NO_3$ leaching, $NH_3$ volatilization or pesticide use [70]. When assessing the effect of a combination of different practices in one single indicator, some complexity is added, since this will also be dependent on the pedoclimatic conditions, the intrinsic performance of the system and the goals set for the sustainability performance [70]. Thus, using such frameworks and interpreting its results should be carried out carefully, since there is a high level of subjectivity that cannot be erased [25].

5.1.4. LADA

LADA is a framework that is focused on the following items: biomass production, yearly biomass increments, soil health, water quality and quantity, biodiversity, economic value of the land use and social services of the land and its use [82]. It is also very solid in providing baseline data for improving the land degradation status, offering valid assets to plant, prioritizing and monitoring [92]. The cost-effectiveness is reasonable, i.e., the mapping activity, which includes the land use systems classification, costs approximately USD 250,000 for a country the size of South Africa [92]. This framework also operates with both local and national scales when assessing the land degradation and sustainable land management, cooperating with different stakeholders and proving applicable in at least 18 countries [93]. This is seen as a strength, since the contribution given by different stakeholders (locally and/or nationally) contributes significantly to equilibrated responses and results. For instance, the same status of a land may be classified differently depending on the stakeholder value system [82]. The LADA framework differs from others in its integration of climate factors, which may account for the long-term performance under climate change conditions.

The use of the framework, however, is still rather limited to people with multi-sectoral expertise [92]. This is linked to the need to build a comprehensive database to store both the quantitative and qualitative data obtained during the assessment operations. The assessment should provide a fixed baseline to monitor future changes and trends, and to feed more in-depth knowledge and understanding into the findings of the national assessment for the area in question [94,95]. Reed et al. [93] also states that, in this framework, land degradation assessment and the impact of the soil management practices that could be applicable in each specific situation should be tighter.

### 5.1.5. SMART

The tool has the advantage of having a high number of indicators to assess the trade-off and synergy analysis. It operationalizes the SAFA guidelines by including indicators that are based on scientific procedures and extensive literature revision. SMART is distinguished from all sustainable assessment frameworks studied by Landert et al. [83] because it integrates the contribution of the stakeholders in its development, which strengthens the acceptance by the end-users while also being specific to local situations [94,96], whereas the others typically involve stakeholders in the application of the framework, but only partly in its development [31]. Therefore, there is a compromise in the intended global applicability of the sustainable assessment tools and the incorporation of a local context.

SMART can be combined with other available tools to improve items such as the system boundary definitions and cut-off criteria when assessing farming activities. The study by Landert et al. [83] used three tools when assessing farm sustainability: COMPAS (an economic farm assessment tool); Cool Farm Tool (a greenhouse gas inventory, water footprint and biodiversity assessment tool, CFT); and the SMART Farm Tool. The results showed that SMART results can be used in combination with quantitative data from COMPAS and CFT. This study was a pioneer in showing the sustainability outcome for 15 different farms in Europe at different stages of their agro-ecological transition. The interdisciplinary tune of this research is characterized by its quantitative contributions and the plurality of view [96]. However, this framework proved to be too time consuming for all of the stakeholders involved, as well as for the interviewers. The combination of SMART with different tools and an improved standard method to incorporate data between the frameworks would facilitate this in the future.

Ssebunya et al. [97] used SMART to assess the sustainability performance of certified organic and fair-trade coffee when compared to non-certified in Uganda. The farm scores were included in the study, which enable analyses of synergies and the trade-off between different sustainable themes. Results showed a link between the certification and the improvement of the sustainable performance of the coffee farms. The framework was also used to enhance the governance objectives by suggesting alterations in group organizations and collective capacities, which, circularly, would also impact other sustainable dimensions. The authors pointed out three main limitations and specific requirements for credible and more consistent outcomes. One of these limitations is related to the comprehensiveness, which is related to the necessary trade-offs for the analysis specificity of some sun-themes. For example, 'Energy Use' and 'Greenhouse Gases' might be more accurately quantified through life cycle assessment methods. Profitability can also be calculated from detailed data from farm incomes and expenditures, whereas this is impossible for other sub-themes. Another limitation is related to the implementation, since the use of SMART requires an adequately trained audit team, involving very time-consuming practice activities to properly understand the functioning of the framework, its indicators and application range. Finally, the team also requires an expertise background on agronomy.

### 5.1.6. PG

PG is a user-friendly tool, with scores of the indicators coming directly from farmers' answers. One of the strengths of this framework is, therefore, its ease of application. On the one hand, data needed to compute the sustainability assessment are easy to obtain from simple interviews with farmers [85], and the questions include accessible data from the farm accounts and management. On the other, the framework was specifically designed to be simple, which means that input data requirements are modest, and are easily translated in the calculation methods and results [84]. This also implies that relatively little time is required for an assessment, since both manuals are simple to use and questions and calculations are easy to follow. This framework was specifically developed for agri-environmental schemes, making it the best option for policy makers wanting to address questions on whether suggested schemes/subsidies will significantly impact the different sustainability

dimensions. Famers also have a direct answer on the impact that future improvements will have on the provision of public goods [84].

Other strengths of PG include the high level of transparency and the opportunity to transform the results directly into understandable outcomes of public goods in agriculture. Additionally, its user friendliness integrates better farmers and provides a useful tool for them to gain awareness on their sustainability farming activities, which is the first step to adopt better practices [96]. The main weaknesses, however, are also related to the simplicity of the tool, based on qualitative data collection and the lack of quantitative indicators, which allows for subjectivity in the scoring and results. Other more minor weaknesses are related to the presence of terminology related to nature conservation, which can be unfamiliar to farmers, the lack of the possibility to select indicators and the impossibility of including indicators within the framework [96].

Scoring the Frameworks

For the environment dimension, RISE, SMART and SAFA show a higher number of indicators covered (seven of eight themes), whereas MASC includes only three themes (water, soil and biodiversity). PG and LADA cover six and four themes, respectively, with water, soil and biodiversity as common themes (Table 5). Although an important subject, climate change seems to be missing in most of the frameworks studied, except in the LADA framework.

In the sociocultural dimension, SAFA is the strongest framework, including nine indicators of a total of twelve themes, followed by SMART covering seven, whereas RISE, MASC and PG cover two themes. SMART and SAFA cover the most important aspects of the sociocultural dimension, whereas RISE assesses only two (workload and wages) and LADA does not assess the sociocultural dimension at the individual level, but rather through land degradation that affects the society as whole (Table 3). In addition to the sociocultural advantages of SMART and SAFA, they enable us to engage stakeholders in different steps in order to increase their acceptance by end-users.

In the economic dimension, SMART, SAFA and PG all cover five themes out of six, followed by RISE and MASC with two themes each (profitability and vulnerability), and LADA with one theme (economic risk). SMART and SAFA assess all themes of the economic dimension except economic risks, whereas PG excludes only the investment theme. Despite the low number of economic themes included, farmers perceive RISE and SMART as the most indicated frameworks for understanding the level of sustainability achieved in their farm because they are based on quantitative data, which are then used for specific contexts [63,64].

In summary, SAFA is the framework with more focus on sociocultural aspects, while still covering some environmental and economic themes. SMART is also homogenous, and covers all three dimensions, but with fewer themes in each one in comparison to SAFA. In contrast, LADA does not include the sociocultural dimension at the individual level and is focused on the environmental dimension. The same is true to some extent for RISE and PG, which include few themes of the sociocultural dimension, while being focused on the environment and /or economy, respectively (Figure 1).

*5.2. Which Frameworks Should Farmers Select?*

To help stakeholders decide which framework is the most suitable for their sustainable assessment, we have developed a decision tree based on possible scales, sectors of applicability and the completeness of sustainability dimensions required (Table 6). For global assessments, there are both SAFA and LADA, but SAFA differs from LADA in assessing food systems in addition to land degradation. In addition, SAFA covers all dimensions, whereas LADA excludes the sociocultural dimension at the individual level, and it includes only a few economic themes.

**Table 5.** Framework's scoring based on their environmental (**A**), sociocultural (**B**) and economic (**C**) assessment themes.

| (A) | Water | Soil | Air | Climate | Plant and Fertility | Biodiversity | Energy Use | Animal Well Being | Total |
|---|---|---|---|---|---|---|---|---|---|
| RISE | X | X | X | - | X | X | X | X | 7 |
| MASC | X | - | - | - | - | X | X | - | 3 |
| LADA | X | X | - | X | - | X | - | - | 4 |
| SMART | X | X | X | - | X | X | X | X | 7 |
| SAFA | X | X | X | - | X | X | X | X | 7 |
| PG | X | X | - | - | X | X | X | X | 6 |

| (B) | Employment Agreement | Workload | Wages | Health Safety | Job Satisfaction | Decent livelihood | Gender Equality | Cultural Diversity | Investment in Local Communities | Employment | Consumer Safety | Transparency | Total |
|---|---|---|---|---|---|---|---|---|---|---|---|---|---|
| RISE | - | X | X | - | - | - | - | - | - | - | - | - | 2 |
| MASC | - | - | - | X | - | - | - | - | - | X | - | - | 2 |
| LADA | - | - | - | - | - | - | - | - | - | - | - | - | 0 |
| SMART | X | - | X | X | - | X | X | X | - | - | - | - | 7 |
| SAFA | X | - | X | X | X | - | X | - | X | X | X | X | 9 |
| PG | - | - | - | - | X | - | - | - | - | X | - | - | 2 |

| (C) | Profitability | Vulnerability | Accountability | Investment | Local Economy | Economic Risk | Total |
|---|---|---|---|---|---|---|---|
| RISE | X | X | - | - | - | - | 2 |
| MASC | X | X | - | - | - | - | 2 |
| LADA | - | - | - | - | - | X | 1 |
| SMART | X | X | X | X | X | - | 5 |
| SAFA | X | X | X | X | X | - | 5 |
| PG | X | X | X | - | X | X | 5 |

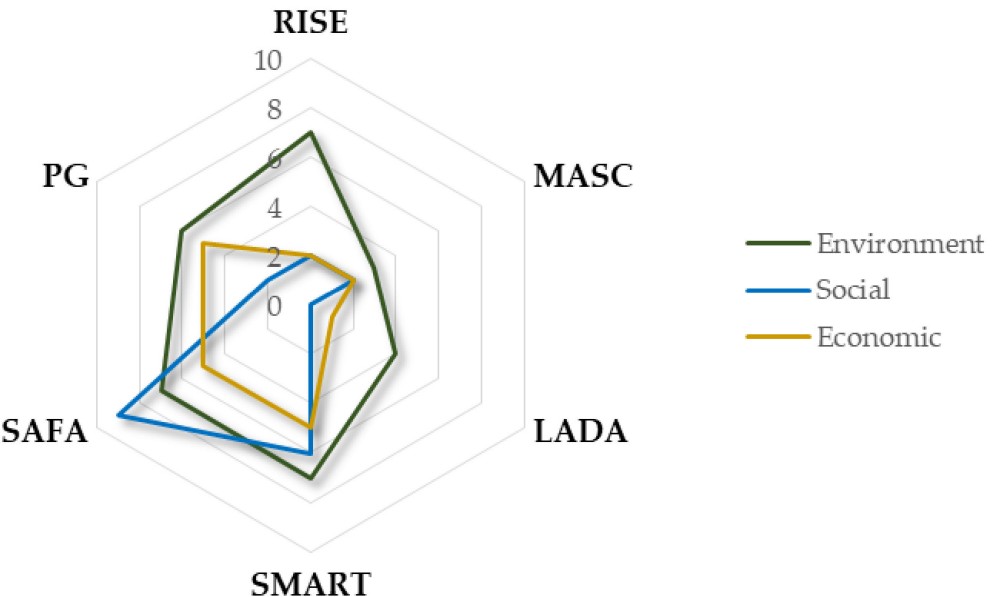

**Figure 1.** Total number of environmental, sociocultural and economic indicators used in each framework under study: RISE, MASC, LADA, SMART, SAFA and PG.

**Table 6.** Decision tree according to the framework scale assessment (global/local), sector of application (cropping system, livestock system, forestry system, urban system and food sector) and completeness of sustainability assessment (environmental, economic and sociocultural dimensions). Icons in black represent a higher number of themes whereas grey represent a lower number of themes in each dimension. Strengths and weaknesses related to the user-friendliness of the tool and the use of qualitative/quantitative data are also mentioned.

| Scale Assessment | Sector of Application | Completeness Assessment | Framework | Strengths (+) and Weaknesses (−) |
|---|---|---|---|---|
| Global | 🌽🐄🌲🏠🏭 | 🌳 $ 👥👥 | SAFA | • Qualitative and quantitative data (+)<br>• Complex framework, requires expert in sustainability (−) |
| | 🌽🐄🌲🏠 | 🌳 $ 👥👥 | LADA | • Qualitative and quantitative data (+)<br>• Limited to people with multi-sectoral expertise (−) |
| Farm | 🌽🐄 | 🌳 $ 👥👥 | RISE | • Quantitative and qualitative data (+)<br>• High number of input data, requires specialist (−) |
| | 🌽🐄 | 🌳 $ 👥👥 | PG | • Only qualitative data used (−)<br>• Scores of the indicators coming directly from farmers answers (+) |
| | 🌽 | 🌳 $ 👥👥 | MASC | • Highly adaptable for qualitative and quantitative data (+)<br>• Requires researcher/professional (−) |
| | 🌽🐄🥤 | 🌳 $ 👥👥 | SMART | • Uses semi-quantitative data (+)<br>• Very time-demanding and limited to scientists (−) |

When the stakeholder intends to perform a sustainability assessment on a farm level, he/she has four choices: RISE, PG, MASC and SMART, but the latest only covers the cropping sector and is rather limited in the number of themes covered. The other three frameworks include cropping and livestock systems, whereas SMART also includes the food sector, which is the only possible choice if that is the user's goal. The selection between RISE, PG and SMART depends on the level of completeness intended for the analysis. SMART covers all dimensions, but with fewer themes in each dimension, whereas the other two include more themes in the environment and economy, respectively. However, MASC and RISE are more complex frameworks, whereas PG is the most user friendly and accessible for farmers.

## 6. Summary and Conclusions

The comparison between the six sustainability assessment frameworks (SAFA, RISE, MASC, LADA, SMART and PG) showed that they have different characteristics with regard to their assessment methodologies, time and data requirements to operate, and different outcomes with a different accuracy and level of complexity. Balancing all of these aspects in the development of the sustainability frameworks in order to meet the expectations of the main actors has proven to be a challenging task.

The high variety of characteristics of each sustainability frameworks derives from the fact that they were developed to serve different end-users: (i) farmers for assessing their farm performance; (ii) advisories and technicians for advising farmers on how they can improve their sustainability; (iii) researchers who conduct comprehensive regional and local assessments adaptable for context-specific conditions by combining, for example, different indicators from different frameworks.

The six sustainability assessment frameworks were compared according to their ability to cover the main themes of environmental, economic and sociocultural dimensions, and their themes were reported. We have also developed a decision tree based on possible scales, sectors of applicability and the completeness of sustainability dimensions required to help stakeholders decide which framework is the most suitable for their sustainable assessment purposes.

This overview study reveals that a multi-actor approach is necessary to enable the acceptance of the outcomes and their adoption by the main actors (i.e., farmers). When a value judgement is incorporated into a framework without involving farmers (e.g., assuming that organic farming will be more sustainable), the results may become irrelevant and are not considered useful by them [58,98,99].

It might be difficult to include alterations occurring in climatic, environmental, socio-economic or technological dimensions, in both the short- and/or long-term in the agricultural and societal aspects, but it may also offer new opportunities for more sustainable development [100]. Therefore, assessing the long-term performance under climate change conditions should be addressed further while assessing agricultural sustainability. For this purpose, realistic climate scenarios should be included.

**Author Contributions:** Conceptualization and writing—original draft preparation, A.A., L.B. and C.S.S.F.; review and editing, R.H. All authors have read and agreed to the published version of the manuscript.

**Funding:** This work is a part of the EU—H2020 project entitled 'Soil Care for profitable and sustainable crop production in Europe' (SoilCare), contract number 677407, which aims to identify and evaluate promising soil improving CS and agronomic techniques, increasing profitability and sustainability across scales in Europe.

**Institutional Review Board Statement:** Not applicable.

**Informed Consent Statement:** Not applicable.

**Data Availability Statement:** Not applicable.

**Acknowledgments:** We would like to thank the handling editor as well as the three reviewers for their constructive comments which considerably improved this paper.

**Conflicts of Interest:** The authors declare no conflict of interest.

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
