# Peer review of "An Overview of Sustainability Assessment Frameworks in Agriculture"

_land, doi:10.3390/land11040537_

Round 1

Reviewer 1 Report

The article submitted to me for review is a review of functioning indicators and frameworks of sustainability assessment in agricultural areas. According to the Authors, the objectives of the manuscript were to: understand the criteria for selecting appropriate frameworks and summarize the range of those used to assess sustainability; identify available frameworks for assessing sustainable agriculture; and analyze the strengths, weaknesses, and applicability of each of the six frameworks identified in the research. All of the objectives have been properly and comprehensively met by the authors. The analyses were based on the whole range of international publications on the subject. The content of the article does not raise my concerns.

However, I have quite serious doubts regarding the type of manuscript adopted by the Authors. The authors have chosen to define their manuscript as: Review. Indeed, the structure, form, and arrangement of the content correspond to this. However, in the instructions for authors there is an important condition for this type of manuscript.  According to the MDPI guidelines, the Review should contain concise and precise information about recent advances in a given field of science. The MDPI framework analyzed in this article can hardly be described as "recent advances in a given field of science." The frameworks identified by the authors are respectively from the years (according to the references) SAFA - 2014, RISE - 2003, MASC - 2009, LADA - 2010, SMART - 2020 and PG - 2009. In fact, only SMART meets the condition indicated earlier. This causes, in my opinion, that the manuscript does not meet the fundamental condition of Review.

I would suggest to the authors, however, that they convert the manuscript into a scientific article. This would obviously require some rewriting of it, but almost all the necessary content and elements can be found in the current version. The analysis of the research to date is very extensive; as a methodology, one can use the definition of the criteria for selecting individual frameworks based on the literature analysis, in the results one can present a comparison of the six selected frameworks and the developed decision tree, in the discussion - indicate the strengths and weaknesses of the individual frameworks. Add to this the suggestion and conclusions of the conducted research and a good scientific article would be created.

Regardless, I have some minor editing comments:

  1. please correct the numbering of the main chapters - all of them have number 1 in the current version?
  2. I understand the idea of placing tables and figures at the end of the paper, but I think they should appear in the text in appropriate places. Please also add their sources.

The article is a good study and I will recommend it for publication after addressing the above comments.

Author Response

Thank you for your valuable comments and suggestions.

We changed the title of the manuscript by deleting "review".

We changed the structure according to the Reviewer's 1 suggestions.

We deleted the "review" throughout the manuscript and replace it by overview or scientific article

We inserted the tables in the appropriate place.

Please refer to the marked and unmarked revised version of the manuscript

Reviewer 2 Report

Land (ISSN 2073-445X)

Manuscript ID: land-1661176

Type: Review

Number of Pages: 31

Title: Assessment of sustainability in agricultural areas: a review of indicators and frameworks.

Dear Authors,

It has been for me a great honour, as well as a pleasantly challenging activity, to review the article entitled “Assessment of sustainability in agricultural areas: a review of indicators and frameworks.”

Overall, the article is interesting and easy to read. It has a good chance of attracting the attention of potential readers. However, I suggest that the Authors introduce a few corrections (given below).

In my opinion, the Introduction chapter well introduces potential readers to the topics discussed by the Authors. The aim of the paper is clearly stated (lines 89-95). This chapter is based on well-chosen literature. However, I would suggest adding a few sentences to broaden the description of the issues related to the risks associated with the loss of biodiversity and the benefits of restoring biodiversity by reintroducing to cultivation primary/primeval plant varieties characterized by high nutritional value (e.g.: https://doi.org/10.3390/agriculture10110556; https://doi.org/10.1016/j.foodres.2010.07.004).

I would also suggest that the Authors show more clearly what the novelty of the submitted article is, what the gap it fills in the existing literature.

Indicators for sustainable agriculture assessment

This chapter is clearly written and supported by well-chosen literature. It is logically divided into the following subchapters and subsections. Minor note: this chapter and the following ones are incorrectly numbered, it should be corrected.

Framework’s for sustainability assessment

In my opinion, the framework is logical and clearly described as well as well illustrated in tables. An error with the numbering of the chapter and subsections crept again.

Synthesis and conclusions (again wrong chapter numbering)

The conclusions are interesting both from a scientific and practical point of view. In my opinion, the added value of the manuscript is the decision tree (based on possible scales, sectors of applicability and the completeness of sustainability dimensions required) developed by the Authors.

I don't feel competent to comment on linguistic correctness as English is not my mother tongue. I can only add that the article is interesting, it reads well and I wish the Authors good luck.

Author Response

Reviewer #2

Number of Pages: 31

Title: Assessment of sustainability in agricultural areas: a review of indicators and frameworks.

Dear Authors,

It has been for me a great honour, as well as a pleasantly challenging activity, to review the article entitled “Assessment of sustainability in agricultural areas: a review of indicators and frameworks.”

Overall, the article is interesting and easy to read. It has a good chance of attracting the attention of potential readers. However, I suggest that the Authors introduce a few corrections (given below).

In my opinion, the Introduction chapter well introduces potential readers to the topics discussed by the Authors. The aim of the paper is clearly stated (lines 89-95). This chapter is based on well-chosen literature. However, I would suggest adding a few sentences to broaden the description of the issues related to the risks associated with the loss of biodiversity and the benefits of restoring biodiversity by reintroducing to cultivation primary/primeval plant varieties characterized by high nutritional value (e.g.: https://doi.org/10.3390/agriculture10110556; https://doi.org/10.1016/j.foodres.2010.07.004).

Reply: I would like to thank the reviewer for this proposition. We think that expanding this issue without discussing the other issues related to land degradation (e.g. erosion, compaction, salinization, etc.) will deviate from the purpose of the manuscript which is to identify the frameworks for sustainability assessment.

I would also suggest that the Authors show more clearly what the novelty of the submitted article is, what the gap it fills in the existing literature.

Reply: It is reported in this version (L81-87, and in the aim of the manuscript, where we reported the decision tree, L88-94).

Indicators for sustainable agriculture assessment

This chapter is clearly written and supported by well-chosen literature. It is logically divided into the following subchapters and subsections. Minor note: this chapter and the following ones are incorrectly numbered, it should be corrected. Reply: Done

Framework’s for sustainability assessment

In my opinion, the framework is logical and clearly described as well as well illustrated in tables. An error with the numbering of the chapter and subsections crept again. Reply: Done

Synthesis and conclusions (again wrong chapter numbering) àdone

The conclusions are interesting both from a scientific and practical point of view. In my opinion, the added value of the manuscript is the decision tree (based on possible scales, sectors of applicability and the completeness of sustainability dimensions required) developed by the Authors.

I don't feel competent to comment on linguistic correctness as English is not my mother tongue. I can only add that the article is interesting, it reads well and I wish the Authors good luck.

Reviewer 3 Report

The topic of the article is interesting. The research design is clear in all sections. Six frameworks have been accurately described. 

The bibliographic search is relevant and thorough with 101 references. A significant part of the bibliographic references are included in the last five years
In my opinion, the article has good scientific soundness
The results are clear.
I would recommend dividing the discussion from the conclusions and making the limitations of the study explicit

Author Response

Dear Editor and dear Reviewers,

We would like to thank all reviewers for their comments and suggestions. They helped improve the quality of the manuscript.

We hope that this version fit your expectations.

Best regards,

Abdallah Alaoui

Reviewer #1

Reply:  Thank you for the constructive and valuable comments that we have addressed. See below.

I would suggest to the authors, however, that they convert the manuscript into a scientific article. This would obviously require some rewriting of it, but almost all the necessary content and elements can be found in the current version.

Reply: We addressed this comment by changing the title (L2-3) and the mention to review article was replaced by Scientific article (L1)

  • The analysis of the research to date is very extensive; as a methodology, one can use the definition of the criteria for selecting individual frameworks based on the literature analysis. Reply: Done, refer to L187
  • in the results, one can present a comparison of the six selected frameworks and the developed decision tree, Reply: Done, refer to L216
  • in the discussion - indicate the strengths and weaknesses of the individual frameworks. Reply: Done, refer to L378
  • Add to this the suggestion and conclusions of the conducted research and a good scientific article would be created. Reply: Done, refer to L634

Minor comments

  • please correct the numbering of the main chapters - all of them have number 1 in the current version? Reply: Done, this confusion was introduced while uploading the file on the journal platform
  • I understand the idea of placing tables and figures at the end of the paper, but I think they should appear in the text in appropriate places. Please also add their sources. Reply: Done, please note the Tables 2-3 can be placed one after the other or placed elsewhere in the text of the Results section

Reviewer #2

Number of Pages: 31

Title: Assessment of sustainability in agricultural areas: a review of indicators and frameworks.

Dear Authors,

It has been for me a great honour, as well as a pleasantly challenging activity, to review the article entitled “Assessment of sustainability in agricultural areas: a review of indicators and frameworks.”

Overall, the article is interesting and easy to read. It has a good chance of attracting the attention of potential readers. However, I suggest that the Authors introduce a few corrections (given below).

In my opinion, the Introduction chapter well introduces potential readers to the topics discussed by the Authors. The aim of the paper is clearly stated (lines 89-95). This chapter is based on well-chosen literature. However, I would suggest adding a few sentences to broaden the description of the issues related to the risks associated with the loss of biodiversity and the benefits of restoring biodiversity by reintroducing to cultivation primary/primeval plant varieties characterized by high nutritional value (e.g.: https://doi.org/10.3390/agriculture10110556; https://doi.org/10.1016/j.foodres.2010.07.004).

Reply: I would like to thank the reviewer for this proposition. We think that expanding this issue without discussing the other issues related to land degradation (e.g. erosion, compaction, salinization, water scarcity, etc.) will deviate from the purpose of the manuscript which is to identify the frameworks for sustainability assessment.

I would also suggest that the Authors show more clearly what the novelty of the submitted article is, what the gap it fills in the existing literature.

Reply: It is reported in this version (L81-87, and in the aim of the manuscript, where we reported the decision tree, L88-94).

Indicators for sustainable agriculture assessment

This chapter is clearly written and supported by well-chosen literature. It is logically divided into the following subchapters and subsections. Minor note: this chapter and the following ones are incorrectly numbered, it should be corrected. Reply: Done

Framework’s for sustainability assessment

In my opinion, the framework is logical and clearly described as well as well illustrated in tables. An error with the numbering of the chapter and subsections crept again. Reply: Done

Synthesis and conclusions (again wrong chapter numbering) Reply: done

The conclusions are interesting both from a scientific and practical point of view. In my opinion, the added value of the manuscript is the decision tree (based on possible scales, sectors of applicability and the completeness of sustainability dimensions required) developed by the Authors.

I don't feel competent to comment on linguistic correctness as English is not my mother tongue. I can only add that the article is interesting, it reads well and I wish the Authors good luck.

Reviewer #3

The topic of the article is interesting. The research design is clear in all sections. Six frameworks have been accurately described. 

The bibliographic search is relevant and thorough with 101 references. A significant part of the bibliographic references are included in the last five years
In my opinion, the article has good scientific soundness
The results are clear.
I would recommend dividing the discussion from the conclusions and making the limitations of the study explicit

Reply: thank you very much for your positive evaluation.

Round 2

Reviewer 1 Report

The authors have considered all my comments and have revised the article as indicated. I accept the current version of the article. A minor correction: further on there are incorrectly numbered chapters (all have No. 1).

Author Response

Dear Reviewer, 

Thank you for your comment. We have fixed now the numbers of the chapters. We already addressed this but, during the submission process, the numbers change to 1. 

I will highlight this to the editor with in an email.

  1. Introduction
  2. General considerations
    • Criteria for selecting sustainability indicators
    • Indicators typically used
  3. Methodology
  4. Results
  5. Discussion
    • Strengths and weaknesses of the frameworks
    • Which frameworks should farmers select?

Reviewer 2 Report

The Authors made necessary corrections and provided convincing explanations. 

Author Response

Dear Reviewer, 

Thank you very much for your time and support.

Best wishes

Abdallah Alaoui

This manuscript is a resubmission of an earlier submission. The following is a list of the peer review reports and author responses from that submission.